# Locating Multiple Sources of Contagion in Complex Networks under the SIR Model

**Xiang Li** [1] 🔘**, Yangyang Liu** [1]**, Chengli Zhao** [1,*]**, Xue Zhang** [1] **and Dongyun Yi** [1,2]

[1] College of Liberal Arts and Sciences, National University of Defense Technology, Changsha 410073, China; lixiang13a@foxmail.com (X.L.); liuyangyang10a@gmail.com (Y.L.); xuezhang@nudt.edu.cn (X.Z.); dongyunyi@nudt.edu.cn (D.Y.)

[2] State Key Laboratory of High Performance Computing, National University of Defense Technology, Changsha 410073, China

[*] Correspondence: chenglizhao@nudt.edu.cn

**Abstract:** Simultaneous outbreaks of contagion are a great threat against human life, resulting in great panic in society. It is urgent for us to find an efficient multiple sources localization method with the aim of studying its pathogenic mechanism and minimizing its harm. However, our ability to locate multiple sources is strictly limited by incomplete information about nodes and the inescapable randomness of the propagation process. In this paper, we present a valid approach, namely the Potential Concentration Label method, which helps locate multiple sources of contagion faster and more accurately in complex networks under the SIR(Susceptible-Infected-Recovered) model. Through label assignment in each node, our aim is to find the nodes with maximal value after several iterations. The experiments demonstrate that the accuracy of our multiple sources localization method is high enough. With the number of sources increasing, the accuracy of our method declines gradually. However, the accuracy remains at a slight fluctuation when average degree and network scale make a change. Moreover, our method still keeps a high multiple sources localization accuracy with noise of various intensities, which shows its strong anti-noise ability. I believe that our method provides a new perspective for accurate and fast multi-sources localization in complex networks.

**Keywords:** multiple sources localization; potential concentration label; anti-noise ability

## 1. Introduction

Contagion propagation [1–3] is an important topic in social network research [4], which brings huge damage to society [5–7], drawing much attention from researchers. When severe contagion outbreaks occur in important places simultaneously, decision-makers including government leaders almost have no ability to deal with the disaster very well. Here is a real case. In 2009, the H1N1 pandemic spread almost simultaneously in Beijing, Shanghai, Fujian and Guangdong provinces, which were the sources propagating the virus in China, and then across the country. After the application of medical treatment over several months, the number of infected people declined gradually and disappeared. If we acquired the infection state of the whole country at an early stage, we could know the cities which spread the virus initially and then take practical actions to prevent it from spreading further. Furthermore, for each originally infected city, we are able to adopt appropriate measures to eliminate it after the disease spread for a short while. Therefore, it makes sense for us to design a pre-warning system, whose core content means a fast and accurate multiple sources of contagion localization method in complex networks under the SIR [8,9] model.

Despite many studies having been done in this field, the problem of multiple sources localization is still a challenging work. On the one hand, the exact number of sources and initial time of sources remain unknown to us; on the other hand, it is inevitable that the randomness of the propagation

process will decrease the accuracy of multiple sources localization methods. In the past ten years, the multiple sources localization problem has attracted many researchers' attention. Shah et al. [10] first proposed a rumor centrality, as a maximum likelihood estimation of single source, in a tree network under the SI model. Then Lou et al. [2] popularized it to a multiple sources localization method. They proposed a Multiple Sources Estimation and Partitioning algorithm, the key to which lies in dividing the network to several disjoint infection regions by infection partitioning, where one region corresponds to one source. Zhu et al. [11] defined Jordan centrality, the farthest distance a node to all the infected nodes and the sources are the nodes with the smallest Jordan centrality. In addition, there is a similar concept, namely distance centrality, the sum distance of the node to the whole infected nodes and the sources have the smallest distance centrality analogously. Fioriti et al. [12] presented to calculate the dynamical age of each node based on the importance of node dynamical. They considered the eigenvalue drop rate of the adjacency matrix as the dynamical age when a node was eliminated from this network and the sources are the nodes with the highest dynamical age. Based on the SIR model and incomplete node information, Zang et al. [13] proposed an advanced unbiased betweenness algorithm. They used a reverse propagation algorithm to build an extended infection graph and marked off several infection subgraphs where we can identify the source with the highest unbiased betweenness. Moreover, based on source identification algorithm, Wang et al. [14] presented label propagation, where they set up an initial label, propagated label and chose the nodes with the tallest label as the sources. Hu et al. [15] combined a backward diffusion-based method with IP to locate both sources and the initial diffusion time with a limited number of observers.

In addition, some scholars study the source localization problems from different angles [16–20]. Nino Antulov-Fantulin et al. [21] proposed a new source localization method based Monte-Carlo simulation under the SIR model. Fu et al. [22] studied a backward diffusion-based source localization method. Based on the times at which the diffusion reached partial observers, the maximum time when the diffusion goes reversely from partial observers to each node is calculated. Then the node with the minimum value is picked up and recognized as the source. Huang et al. [23] used observers to diffuse reversely and found the node as the source with minimum variance yields, resolving the single source localization problem in the temporal network.

In this paper, we propose the Potential Concentration Label method (PCL) to locate multiple sources of contagion in complex networks under the SIR model. The main idea in this paper reflects that the sources prefer to exist in the infection region with more infected neighbor nodes where the nodes have the maximal value of potential concentration label just right. In the following sections, we first define the Potential Concentration Label and propose the PCL method. After that, we test the performance network parameters on sources localization accuracy in synthetic networks and real networks, compared to the other four benchmark methods. Finally, some experiments are carried out to measure the anti-noise ability of our method.

## 2. Model and Method

### 2.1. SIR Model for Contagion Propagation

In this work, we focus on an undirected graph $G = \{V, E\}$, where $V$ is the set of nodes and $E$ is the set of edges. Each node $v \in V$ has its possible state—Susceptible ($S$), Infected ($I$), Recovered ($R$). The susceptible nodes represent the people who are infected easily but have not been infected yet, meanwhile the infected nodes denote the citizens who have already been infected and are capable of infecting other nodes. The recovered nodes are the individuals who remain immune or die. Suppose that there is a time-slotted system. At first, only several nodes are infected, which are the contagion sources in the network. Meanwhile, the other nodes are susceptible. At each time step, each infected node infects its susceptible neighbors with probability $p$ independently, that is, a susceptible node is infected with probability $1 - (1 - p)^n$ when it has $n$ infected neighbors. Meanwhile, the infected nodes

turn to be recovered with probability $q$. Additionally, the recovered nodes will not be infected, which may die or be removed.

### 2.2. Problem Formulation

As a contagion propagates through a complex network under the SIR model, all the nodes will change infection state as time goes by. The susceptible nodes may be infected by infected neighbour nodes and the infected nodes recover to a recovered node with a certain probability. Due to the emergency response to contagion, we mainly consider an initial infection situation of the whole network and only collect two states, infected and uninfected (susceptible, recovered), of all nodes. Accordingly, the problem of the multiple sources localization problem can be described as—given the simple snapshot of the network at an early certain moment, how can we accurately locate multiple sources?

It is common that we know the state of almost all nodes, but we have no ability to distinguish the susceptible nodes from the recovered nodes. Therefore, all nodes can be divided into two states—infected and uninfected, which decreases the accuracy of multiple sources localization certainty.

### 2.3. Potential Concentration Label Definition

In the early period of severe contagion propagation, disease outbreaks through a crowd quickly. It comes to the situation that the nodes around sources are more likely to be the infected nodes, that is, the sources are surrounded by many infected nodes. Only by depending on the infection states can we locate the sources in a complex network accurately.

Inspired by Figure 1a, which shows the concentration of a pollutant, it is clear that the sources are more likely to be the node set $\{d, k\}$, whose concentration is the highest (10). In fact, to get the state of each node is not easy, for example, some sensors do not have the capacity to measure concentrations, and can only judge whether the concentration surpasses a threshold value or not, and even then we may lose the concentration information. Therefore, the information we can obtain is incomplete, just like in Figure 1b. We can see two concentration states easily, 0 or 1 (1 denotes concentration over 8, 0 denotes concentration under 8) in a network, where an error occurred with node $c$. It seems we have no ability to identify the sources according to these concentrations, which is similar to the infection situation of contagion.

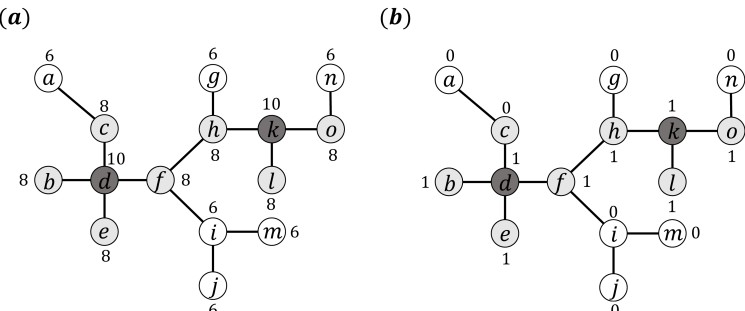

**Figure 1.** A snapshot of the pollutant diffusion process. The letter denotes the node in network and the digital represents the concentration label of each node. The propagation sources are $S^\star = \{d, k\}$. (**a**) pollutant concentration of network diagram. (**b**) incomplete pollutant concentration of network diagram, which is similar to the contagion situation of the network.

Therefore, a new index needs to be proposed so as to distinguish between the sources and other nodes for incomplete pollutant diffusion and contagion propagation. We think the node with more infected neighbors, including the first order neighbor, the second order neighbor and so forth, is closer to the sources. Based on the above analysis, we propose a new concept, namely a potential concentration label, denoted by $\mathscr{L}$. The potential concentration label is determined by its initial label

and the labels of neighbor nodes. The experiments demonstrate that it is a good index for locating multiple sources of contagion in complex networks under the SIR model.

### *2.4. The* PCL *Method*

In this section, we present the PCL method at length in this section. The purpose of PCL is to locate multiple contagion sources, which is realized by following four steps in Algorithm 1.

### 2.4.1. Step 1: Label Assignment in the Snapshot of Network

Due to the incomplete information, only two states can be seen in the network—infected and uninfected (susceptible and recovered). The infection state of nodes **X** is shown as follows—infected nodes carry the virus, denoted by 1; uninfected nodes carry no virus, denoted by 0. That is, if node $i$ is infected, then $\mathscr{L}_i^0 = 1$; if node $i$ is uninfected, then $\mathscr{L}_i^0 = 0$, where $\mathscr{L}_i^0$ is the initial potential concentration label of node $i$.

### 2.4.2. Step 2: Adding One Hub Node to the Network

In real networks, it comes up all the time that the network we acquire is disconnected, but connected actually. To avoid this situation, we can add a hub node in the network, which has a link with every node, to make it connect for certain and to increase its connectivity. Besides, the possibility of this node being infected is high enough that we assign label 1 to it directly.

### 2.4.3. Step 3: Potential Concentration Label Calculation by Iteration

The potential concentration label of a node is connected with the potential concentration label of neighbor nodes and its initial potential concentration label, so the potential concentration label of node $i$ at $t$ iterations becomes:

$$\mathscr{L}_i^t = \alpha \sum_{j \in \Gamma_i} \mathscr{T}_{i,j} \mathscr{L}_j^{t-1} + \beta \mathscr{L}_i^0 \tag{1}$$

where $\alpha, \beta$ is the proportionality coefficient, $\Gamma_i$ represents the first order neighbors of node $i$.

Before starting iteration, we should build an adjacency matrix $A$ and a degree matrix $D$. Matrix $A$ is decided by edge $E$, where $A_{ij} = 1$ represents node $i$ and node $j$ have an edge. Matrix $D$ is a diagonal matrix, where the $i$-th element is the sum of $i$-th row of matrix $A$. The transmission probability matrix $\mathscr{T}$ from neighbors is decided by adjacency matrix $A$ and degree matrix $D$, such that $\mathscr{T} = D^{-\frac{1}{2}} A D^{-\frac{1}{2}}$.

The state of a node at moment $t$ is mainly dependent on the states of its neighbor nodes at moment $t - 1$. Apparently, the potential concentration label of each node at moment $t$ is proportional to the initial potential concentration label. Therefore, we choose $\alpha > 0, \beta > 0$.

Thanks to the hub node, the diameter of the network decreases to two. That is to say, every node only has the first order neighbor and the second order neighbor. Therefore, a node acquires the label information from other nodes in the network, only requiring two iterations. It spends little time in getting the potential concentration label.

---

**Algorithm 1** Potential Concentration Label

---

**Input:** The network topology $G$ and infection state $\mathbf{X}$.
**Output:** The multiple sources $S^\star$.

1: Set up the initial label $\mathscr{L}_i^0, i \in V$;
2: Add a hub node to the network and $\mathscr{L}_{N+1}^0 = 1$;
3: Construct the transmission matrix $\mathscr{T} = D^{-\frac{1}{2}} A D^{-\frac{1}{2}}$
4: **for** t=1:$t1$ **do**
5: $\quad \mathscr{L}_i^t = \alpha \sum_{j \in \Gamma_i} \mathscr{T}_{i,j} \mathscr{L}_j^{t-1} + \beta \mathscr{L}_i^0$
6: **end for**
7: $\alpha > 0, \beta > 0, \Gamma_i$ represents the first order neighbors of node $i$.
8: We choose the nodes with maximal value as the sources $S^\star$;
9: **return** $S^\star$.

---

### 2.4.4. Step 4: The Multiple Sources Localization

The central idea of this paper is that the sources prefer to exist in an infection region with more infected nodes, meanwhile the potential concentration label of sources is superior to that of neighbor nodes. After several iterations, there are several maximal values of potential concentration labels existing in the network. Finally, we choose the nodes with the maximal value as multiple sources.

### 2.5. A Simple Example of Multiple Sources Localization

To better describe the PCL method, we just introduce a simple example of multiple sources localization. Given a snapshot of the network at some point, we can know the infection state of all the nodes. In addition, the sources are $\{f, h\}$. From Figure 2, it is easy to find that the node $f$ and $h$ always have the maximal value. According to PCL, we see nodes $\{f, h\}$ as the estimated sources, which also are the true sources.

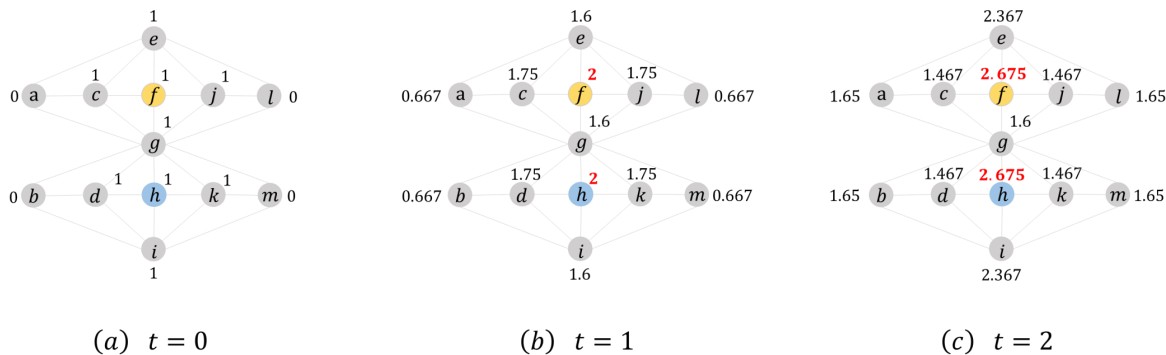

(a) $\quad t = 0$        (b) $\quad t = 1$        (c) $\quad t = 2$

**Figure 2.** An example of multiple sources localization. (**a**–**c**) represent the potential concentration label of the network at different iteration frequencies.

## 3. Simulation and Analysis

### 3.1. Data Descriptions and Measurements

To evaluate the performance of PCL method, we firstly introduce several synthetic networks, that is, ER, WS and BA, and real networks, that is, Karate, Lesmis, Adjnoun, Football, Jazz and USAir, as the experimental data. Synthetic networks are controllable, where network parameters can be adjusted, so that many tests can be done to verify the efficiency of the method. What is more, the data of Karate, Lesmis, Adjnoun and Football networks can be downloaded via the network data of Newman [24]. The other data sets come from the corresponding references. Basic characteristics are shown in Table 1.

**Table 1.** The parameter of real network. The quantities $N$, $|E|$, $< k >$ represent the number of nodes, the number of edges and the average degree, respectively.

| Network | $N$ | $|E|$ | $< k >$ | Description |
|---------|-----|-------|---------|-------------|
| Karate | 34 | 78 | 4.59 | The friendships between 34 members of a karate club at a US university in the 1970s |
| Lesmis | 77 | 258 | 6.70 | The characters relationships in the novel Les Miserables |
| Adjnoun | 112 | 425 | 7.59 | The common adjectives and nouns in the novel David Copperfield by Charles Dickens |
| Football | 115 | 613 | 10.66 | The games among 115 American college football teams |
| Jazz [14] | 198 | 2742 | 27.70 | The jazz bands attending performance from 1912 to 1940 |
| USAir [15] | 332 | 2126 | 12.81 | The flights in the USA |

As we know, F-measure is usually used to check the accuracy of estimated or identified sources in a complex network [25]. It can be defined as follows:

$$F \ measure = \frac{2 * precision * recall}{precision + recall} \tag{2}$$

where precision is the ratio of the number of correctly identified sources over the number of all retrieved sources which is defined in Equation (3) and recall is the ratio of the number of correctly identified sources over the ground truth source, defined in Equation (4)

$$precision = \frac{|\{retrieved \ sources\} \cap \{true \ sources\}|}{|\{retrieved \ sources\}|} \tag{3}$$

$$recall = \frac{|\{retrieved \ sources\} \cap \{true \ sources\}|}{|\{true \ sources\}|} \tag{4}$$

In this paper, suppose that we already know the number of sources so that retrieved sources equal true sources, that is, precision equals F-measure.

Therefore, we choose the precision as the evaluation index of sources localization accuracy in this paper. The situation we face is a serious contagion so that we suppose infection probability $p = 0.8$, recovery probability $q = 0.1$. What is more, the results are obtained by averaging over 100 independent realizations.

*3.2. Optimal Iteration Frequency Choice*

We choose the nodes with the maximal value of the potential concentration label as the sources and the potential concentration label is related to the number of iterations. Therefore, we next test the performance of our multiple sources localization method under six iteration frequencies in synthetic networks and real networks, which can help us find the appropriate iteration frequency.

Figure 3 shows that the source's localization accuracy changes sharply when $t1$ is different. It is an interesting phenomenon that the accuracy reaches its highest when $t1 = 2$. The hub node plays a decisive role in the change of accuracy. On the first iteration, the hub node is an unnecessary node which brings error to the potential concentration label of each node. However, on the next iterations, the hub node transmits all node labels to each node as a bridge, which increases the accuracy of sources localization. Moreover, there is a turning point when $\beta = 0$. The accuracy is higher for $\beta < 0$ than it is for $\beta > 0$. The main reason lies in the incomplete infection information where a recovered node has actually been infecte , especially the sources, but it is considered to be uninfected when calculating. To get a better multiple sources localization performance, we choose $t1 = 2, \beta = -1$ in the following experiments.

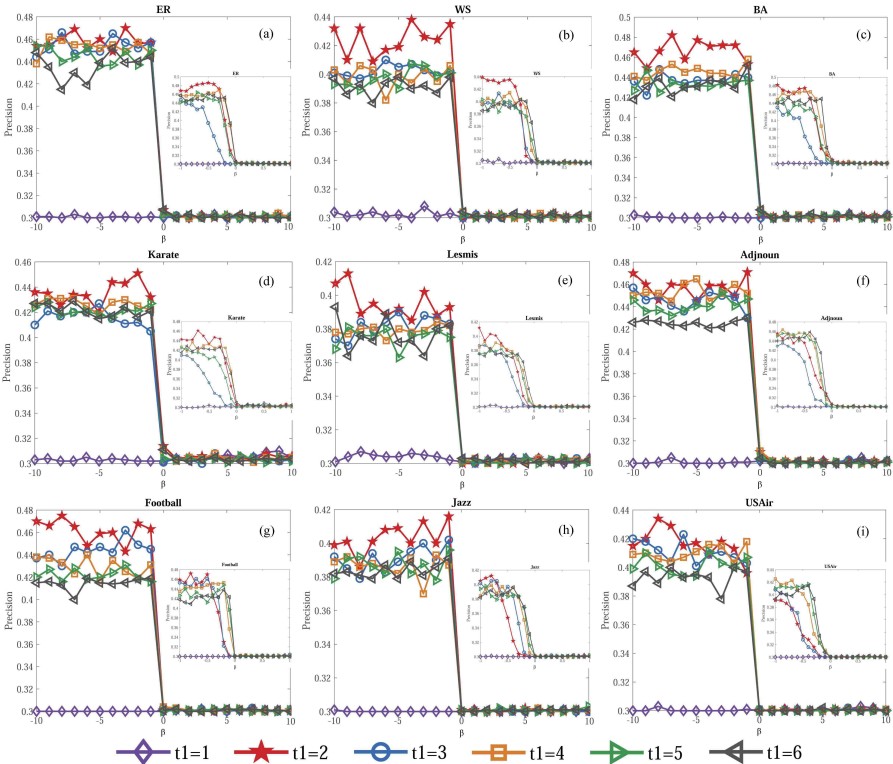

**Figure 3.** Sources localization accuracy with various frequencies of iteration in nine networks. (**a–i**) The relationship between *precision* and $\beta$ in synthetic networks and real networks. Without loss of generality, we suppose $\alpha = 1$. $t1$ denotes the number of iterations.

### 3.3. Comparison Methods

To compare with the performance of PCL, we pick up some sources localization methods as benchmarks.

Distance Centrality (DC) [10]—represents the sum of the distances from one node to all the infected nodes. The sources usually have the smallest Distance Centrality.

Jordan Centrality (JC) [11]—denotes the maximum of the distances from one node to all the infected nodes. The sources prefer to have the least Jordan Centrality.

Unbiased Betweenness Centrality (UBC) [13]—the betweenness of one node eliminates the effect of degree, namely unbiased betweenness. The nodes are the sources, which always have the biggest unbiased betweenness.

Modified Label Propagation based Source Identification (LPSI) [14]: This method lets infection status iteratively propagate in the network as labels, and finally uses local peaks of the label propagation result as source nodes.

### 3.4. Sources Localization in Synthetic Networks

To test the efficiency of the PCL method, we first carry out some experiments in synthetic networks, that is, the Radom (ER) network [26], the Watts-Strogtz small world(WS) network [27], and the scale-free (BA) network [28]. The ER network and WS network are homogeneous networks, and the BA network is a heterogeneous network. We focus our attention on the influence the network parameter has on sources localization accuracy. The main parameters are the scale of network $N$, average degree $<k>$ and the number of sources $s$.

This paper mainly considers the sources localization problem, there is no denying that the number of sources is the most important network parameter. At first, we examine the effects the number of sources has on the performance of sources localization. Figure 4 shows that when the number of sources increases, the sources localization accuracy has a decrease tendency for all the methods, that is,

PCL, LPSI, DC, JC and UBC. When the number of sources becomes large, multiple sources may be too closed to identify them easily. To find the number of sources accurately is the first problem we need to solve urgently. In a short, in the above three synthetic networks, PCL behaves better than the other four methods in sources localization accuracy. With the increasing of the number of sources, the sources localization accuracy of PCL only declines slightly, reflecting its strong robustness.

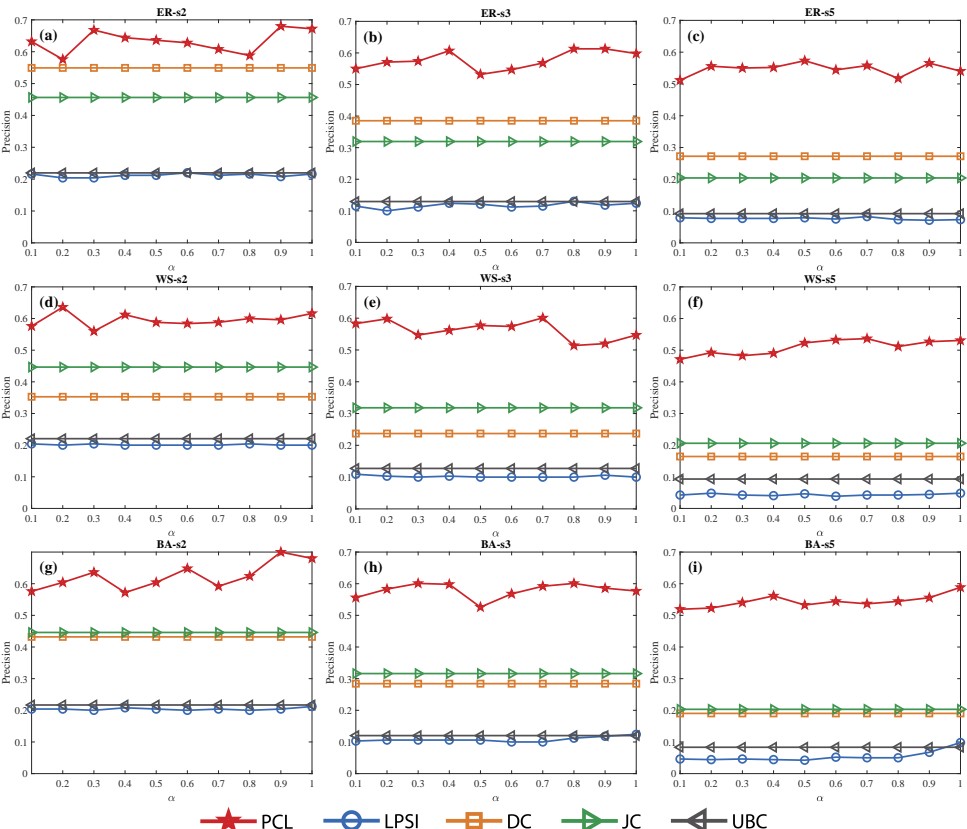

**Figure 4.** Sources localization accuracy of different numbers of sources with five methods in synthetic networks. (**a**–**c**) The relationship between *precision* and $\alpha(\beta = -1)$ for different number of sources in ER network. The number of sources is $s = 2, 3, 5$ respectively, which is similar to Figure 5. The scale of the network is $N = 100$ and the average degree is $< k >= 4$. (**d**–**f**) The difference is that the experiments are carried out in the WS network. (**g**–**i**) Similarly, tests are done in the BA network. Besides, all contrast experiments (including subsequent experiments) involve the above five methods, that is, PCL, LPSI, DC, JC and UBC, which is represented by five different colors.

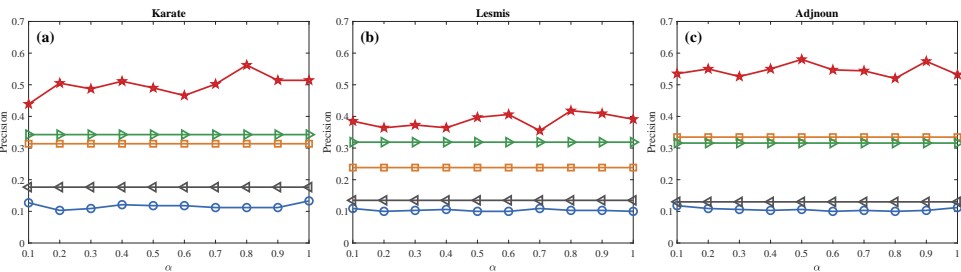

**Figure 5.** *Cont.*

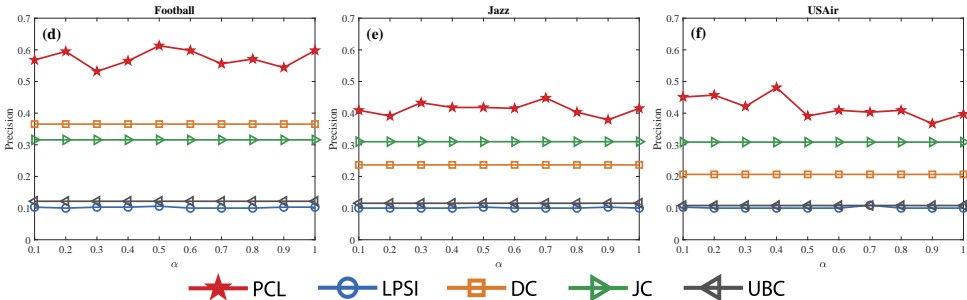

**Figure 5.** The relationship between sources localization precision and $\alpha(\beta = -1)$ with five methods in real networks. (**a–f**) represent the multiple sources localization performance in Karate, Lesmis, Adjnoun, Football, Jazz and USAir network respectively. The number of sources is $s = 3$ in all networks.

From Figure 6, we find that the average degree has little influence on the sources localization accuracy with almost all methods. The results of the four methods in the ER network distinguish that PCL > DC > JC > UBC > LPSI; meanwhile those in the WS and BA networks distinguish that PCL > JC > DC > UBC > LPSI. All in all, PCL can always solve the sources localization problems, no matter whether the network is sparse or not. In other words, the accuracy of PCL keeps very robust when the number of edges in the network increases or decreases.

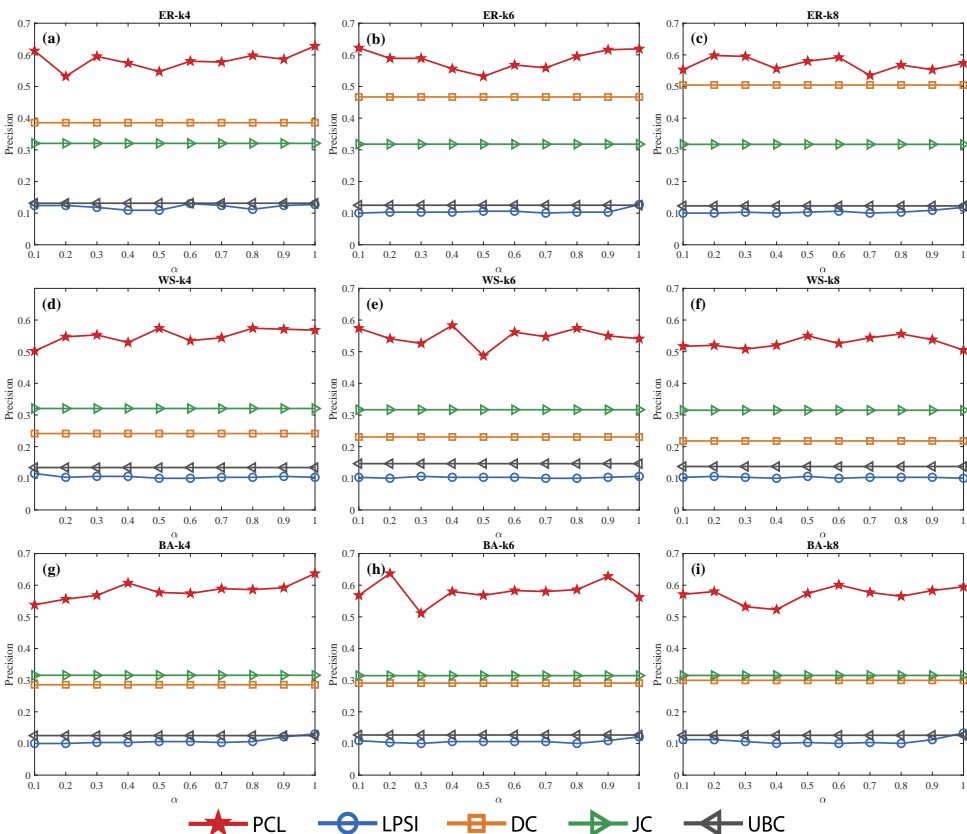

**Figure 6.** Sources localization accuracy of different average degree with five methods in synthetic networks. (**a–c**) Results in ER network. The average degree is $< k >= 4, 6, 8$ respectively, the scale of network is $N = 100$. The number of sources is $s = 3$, the same as Figure 7. (**d–f**) Results in WS network. (**g–i**) Results in BA network.

For different scales of networks, Figure 7 indicates that the sources localization accuracy has a mild fluctuation with the increasing of network scale for all the methods except for DC. In the ER network, the DC method can get a higher accuracy when the scale of network increases. Of course, the

accuracy of PCL keeps robust when network size changes. Thanks to its result, we can generalize this method to large networks based on the background of big data.

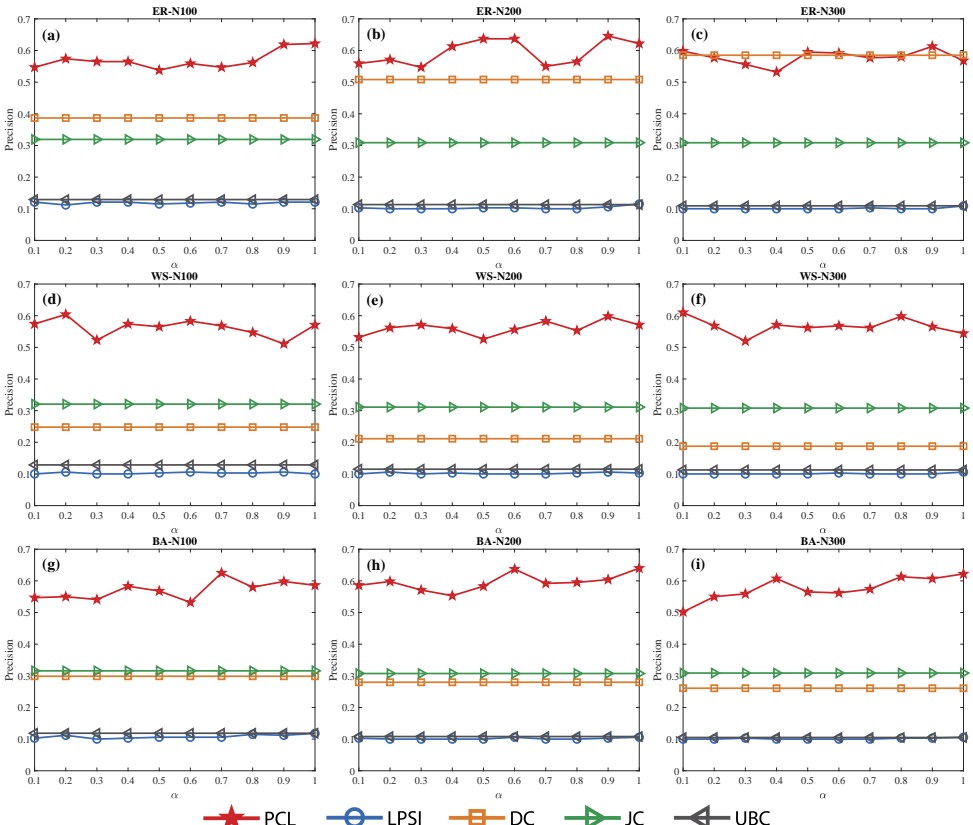

**Figure 7.** Sources localization accuracy of different network scale with five methods in synthetic networks. (**a**–**c**) Results in ER network. The scale of the network is $N = 100, 200, 300$ respectively, the average degree is $< k >= 4$. (**d**–**f**) Results in WS network. (**g**–**i**) Results in BA network.

### 3.5. Sources Localization in Real Networks

In addition to the synthetic network, we test the performance of the above five methods in real networks (Karate, Lesmis, Adjnoun, Football, Jazz and USAir). These networks are social networks, where propagation usually occurs.

From Figure 5, we can find that PCL has the highest sources localization accuracy in all real networks. Meanwhile, the sources localization accuracy of PCL keeps robust with a different network structure. Moreover, it confirms that the average degree and the scale of the network have less effect on sources localization accuracy. All in all, PCL behaves best in sources localization accuracy of five different methods, that is, PCL, LPSI, DC, JC and UBC.

## 4. Anti-Noise

An efficient method needs to keep high accuracy under noise of various intensities. In this section, noise disposal strategies and infection state noise are taken into account so as to test the anti-noise of the sources localization method.

It is very common that the infection information of partial nodes may be lost. Now suppose that there is 20% nodes in a network unknown to us. There are three strategies to deal with it. (*i*) All-inf, a strategy where the nodes without infection information are considered to be infected; (*ii*) None-inf, a strategy where the nodes without infection information are regarded as uninfected; (*iii*) Rand-inf, a strategy where the nodes without infection information are thought to be infected randomly. Next, we

test the performance of sources localization accuracy with five methods in synthetic networks and real networks. The results are shown in Tables 2 and 3.

**Table 2.** The average precision (PCL vs. LPSI vs. DC vs. JC vs. UBC), $\alpha = 0.1 : 0.1 : 1, \beta = -1$, for different strategies of dealing with unknown infection information in synthetic networks and real networks with 20% uncertainty. The number of sources is $s = 3$ in all networks, $< k > = 4$, $N = 100$ in ER, WS and BA network, the connect probability is 0.1 in WS network. Besides, a bold number denotes the highest accuracy of sources localization in each network.

| Network | Strategy | PCL | LPSI | DC | JC | UBC |
|---------|----------|------|------|------|------|------|
| **ER** | All-inf | 53.17 | 11.77 | 24.31 | 30.84 | 12.31 |
| | None-inf | **58.60** | 12.22 | 25.27 | 30.97 | 12.31 |
| | Rand-inf | 55.51 | 12.16 | 25.60 | 31.01 | 13.36 |
| **WS** | All-inf | 47.59 | 10.24 | 18.76 | 31.12 | 12.94 |
| | None-inf | **54.28** | 10.21 | 18.16 | 31.12 | 12.55 |
| | Rand-inf | 50.11 | 10.21 | 17.92 | 30.87 | 13.09 |
| **BA** | All-inf | 51.16 | 10.87 | 19.42 | 30.89 | 12.10 |
| | None-inf | **57.31** | 10.81 | 18.88 | 30.93 | 11.98 |
| | Rand-inf | 54.25 | 10.87 | 18.79 | 30.91 | 12.07 |
| **Karate** | All-inf | 46.27 | 11.83 | 27.67 | 33.09 | 16.48 |
| | None-inf | **50.05** | 11.77 | 27.85 | 33.18 | 17.47 |
| | Rand-inf | 48.22 | 11.95 | 28.39 | 33.22 | 16.30 |
| **Lesmis** | All-inf | 36.31 | 10.24 | 19.18 | 31.10 | 14.02 |
| | None-inf | **36.58** | 10.69 | 19.54 | 31.38 | 13.60 |
| | Rand-inf | 36.10 | 10.51 | 18.97 | 31.41 | 14.44 |
| **Adjnoun** | All-inf | 53.08 | 10.69 | 22.06 | 30.81 | 12.97 |
| | None-inf | **53.56** | 10.48 | 22.57 | 30.96 | 12.19 |
| | Rand-inf | 52.24 | 10.30 | 21.52 | 30.69 | 12.13 |
| **Football** | All-inf | 51.13 | 10.15 | 31.09 | 30.72 | 12.25 |
| | None-inf | **59.92** | 10.27 | 30.79 | 30.79 | 11.95 |
| | Rand-inf | 52.63 | 10.21 | 31.42 | 30.83 | 12.37 |
| **Jazz** | All-inf | 38.53 | 10.24 | 19.42 | 30.34 | 11.35 |
| | None-inf | **41.62** | 10.12 | 18.13 | 30.56 | 11.50 |
| | Rand-inf | 38.59 | 10.12 | 19.03 | 30.55 | 11.35 |
| **USAir** | All-inf | 37.90 | 10.12 | 13.93 | 30.38 | 11.08 |
| | None-inf | 36.04 | 10.21 | 13.45 | 30.30 | 10.90 |
| | Rand-inf | **38.47** | 10.06 | 13.66 | 30.27 | 11.05 |

Table 2 suggests that the sources localization accuracy of PCL reaches its highest among all the methods with each strategy in each network. The accuracy changes slightly, due to different strategies of dealing with noise, of the whole methods except for PCL. In most cases, when noise exists, PCL method achieves the highest sources localization accuracy, mostly choosing the None-inf strategy to deal with noise.

Except for the different strategies for dealing with unknown infection information, we further study the sources localization performance under noise of three various intensities (*ni*), which denotes the proportion of nodes we are unaware of. The noise intensities are shown such that $ni = 0.05$, $ni = 0.1, ni = 0.2$. From Table 3, with increasing noise intensity, the sources localization accuracy of all methods decreases in all networks. Apparently, our method shows a huge advantage in sources localization in all methods. PCL achieves the highest sources localization accuracy not only in an ideal situation (without noise), but also in a real situation(with noise).

**Table 3.** The performance of sources localization accuracy with PCL vs. LPSI vs. DC vs. JC vs. UBC in synthetic networks and real networks under noise of various intensities. The *precision* is the mean value when $\alpha = 0.1:0.1:1$. The number of sources is $s = 3$ in all networks, $< k >= 4$, $N = 100$ in the ER, WS and BA networks, the connect probability is 0.1 in the WS network. Moreover, the strategy of dealing with unknown infection information chooses Rand-inf.

| Network | Noise | PCL | LPSI | DC | JC | UBC |
|---------|-------|-----|------|-----|-----|-----|
| **ER** | 0.05 | 56.83 | 11.68 | 30.07 | 31.07 | 12.43 |
| | 0.10 | 54.73 | 12.34 | 26.05 | 31.09 | 12.70 |
| | 0.20 | 51.61 | 11.50 | 19.30 | 30.92 | 12.58 |
| **WS** | 0.05 | 52.81 | 10.24 | 20.47 | 31.24 | 13.09 |
| | 0.10 | 50.65 | 10.33 | 18.52 | 31.01 | 13.12 |
| | 0.20 | 45.19 | 10.36 | 15.22 | 30.96 | 13.12 |
| **BA** | 0.05 | 56.17 | 10.60 | 21.16 | 30.78 | 12.31 |
| | 0.10 | 54.13 | 10.54 | 18.64 | 30.83 | 12.91 |
| | 0.20 | 49.78 | 10.45 | 15.97 | 30.84 | 12.28 |
| **Karate** | 0.05 | 49.45 | 11.50 | 28.12 | 33.34 | 16.69 |
| | 0.10 | 48.01 | 11.38 | 28.15 | 33.39 | 17.44 |
| | 0.20 | 43.87 | 11.74 | 25.66 | 32.82 | 15.73 |
| **Lesmis** | 0.05 | 36.91 | 10.33 | 20.50 | 31.43 | 13.87 |
| | 0.10 | 36.61 | 10.51 | 19.69 | 31.41 | 13.99 |
| | 0.20 | 33.85 | 10.69 | 17.23 | 31.17 | 13.90 |
| **Adjnoun** | 0.05 | 54.46 | 10.48 | 26.44 | 30.94 | 12.34 |
| | 0.10 | 52.12 | 10.45 | 21.46 | 30.83 | 12.01 |
| | 0.20 | 47.89 | 10.45 | 17.74 | 30.81 | 12.13 |
| **Football** | 0.05 | 55.75 | 10.24 | 34.24 | 30.73 | 12.34 |
| | 0.10 | 52.60 | 10.24 | 31.12 | 30.86 | 12.46 |
| | 0.20 | 49.69 | 10.27 | 27.40 | 30.95 | 12.22 |
| **Jazz** | 0.05 | 40.27 | 10.15 | 21.61 | 30.63 | 11.17 |
| | 0.10 | 40.84 | 10.18 | 18.43 | 30.36 | 11.08 |
| | 0.20 | 36.43 | 10.12 | 16.21 | 30.44 | 11.38 |
| **USAir** | 0.05 | 40.09 | 10.06 | 14.83 | 30.23 | 11.05 |
| | 0.10 | 39.52 | 10.18 | 14.26 | 30.22 | 10.66 |
| | 0.20 | 36.37 | 10.24 | 13.27 | 30.30 | 10.75 |

## 5. Conclusions and Discussion

In this paper, we study multiple sources of the contagion localization problem under the SIR model. Given the snapshot of a network, we propose a fast and more accurate multiple sources localization method, namely Potential Concentration Label. What matters in this method is to find the nodes with the maximal value of the potential concentration label as the sources. Firstly, we assign the initial concentration label to each node according to its infection state; next, it begins the label propagation process, where the label of one node is determined by its neighbors' and its initial own, through two iterations; finally, we choose the nodes with the maximal value of the potential concentration label as the contagion sources. The experiments demonstrate that when the number of sources increases, the sources localization accuracy of our method decreases gradually. However, it keeps very robust as the average degree and network scale make a change. Compared to other benchmark methods, this method has a low time complexity and higher sources localization accuracy in synthetic networks and real networks. What is more, the anti-noise ability of our method is strong enough, which shows its effectiveness.

Although our method provide a new reference for the problem of multiple sources localization in complex networks, much work still needs to be done. The issue of sources localization we proposed is

based on an undirected network, while this method may extend to the directed network. Moreover, the network topology we consider in this paper is static and known to us. In fact, the topology in the real world remains dynamic as time goes by. This makes us improve our model so as to solve the sources localization problem in a temporal network [29,30]. In addition, as we know, few theoretical and practical studies have focused attention on multiple sources localization in multi-layer networks [31].

**Author Contributions:** Conceptualization, D.Y.; methodology, X.L.; software, Y.L.; validation, Y.L.; formal analysis, Y.L.; investigation, X.Z.; resources, X.L.; data curation, X.L.; writing—original draft preparation, X.L.; writing—review and editing, X.Z.; visualization, Y.L.; supervision, X.L.; project administration, C.Z.; funding acquisition, C.Z. This paper was prepared the contributions of all authors. All authors have read and approved the final manuscript.

**Funding:** This work is supported by the National Key R&D Program of China (Grant No. 2017YCF1200301).

**Acknowledgments:** We thank Xiaojie Wang, Caixia Yu and Qiangjuan Huang for valuable discussions.

**Conflicts of Interest:** The authors declare no conflict of interest.

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
