# Peer review of "Locating Multiple Sources of Contagion in Complex Networks under the SIR Model"

_applsci, doi:10.3390/app9204472_

Round 1
Reviewer 1 Report
The manuscript presents a really interesting and exhaustive investigation of locating multiple sources of contagion in complex networks, through the approach namely Potential Concentration Label Method.
#The structure of the manuscript is really clear and all the sections are presented in a good way.
#The manuscript is well-written, clear and concise, and English is comprehensible and satisfactory.
#The model and the methodology referred to the approach and the comparison analysis with other benchmark methods, result in a pretty easy one and simulation and the test appear correct.
#The figures are clear and relevant with adequate captions.
Although I have few minor comments below that the authors should address to improve their study, the work can be considered for publication in Applied Science.
Below some considerations and comments:
#The abstract section is well written but I suggest, at the end of section, to shed light the findings of the paper, in a synthetic way, and the real contribution applying the PCL approach. The sentence “…method behaves best compared with four benchmark methods..” it's too general.
#The introduction is too brief and it would be interesting to introduce aspects concerning real cases of multiple sources of contagion, with some references. It would be interesting too, underline other factors such as collective awareness and social relationships in epidemic spreading, etc. There is a lot of scientific literature regarding that. Moreover, I suggest highlighting the problems of decision making, health management in the case of simultaneous infections in real cases. This paper should better suggest how people in the health-care field and management in emergency or risky events can benefit from such an approach.
#The author chooses to explore this problem in complex networks under th SIR model. I suggest extending the SIR assumption discussion. In section 2.2 there is not a clear examination of the impact of assuming a SIR propagation model. The author identifies two node states in contrast with the SIR assumption of the three states.
#I suggest adding some other references more recent.
This paper presents a model and the analysis methods that are based on scientific literature and although it is not strongly innovative, it introduces some interesting aspects and for this reason, in my opinion, it is suitable for the publication.
Reviewer 2 Report
See attached file.

Round 2
Reviewer 2 Report
The paper has been improved related to the former version. It contains original results which are illustrated with examples. The theoretical and formal technical support developed in the article is also well-worked and adequate.